



# Trends in long-term hydrological data from European karst areas: insights for groundwater recharge evaluation

Markus Giese[1,2], Yvan Caballero[1,2], Andreas Hartmann[3], Jean-Baptiste Charlier[1,2]

[1] BRGM, Univ. Montpellier, Montpellier, France
[2] G-eau, UMR 183, INRAE, CIRAD, IRD, AgroParisTech, Supagro, BRGM, Montpellier, France
[3] Institute for Groundwater Management, TU Dresden, Dresden, Germany

*Correspondence to*: Markus Giese (markus.giese@gu.se)

**Abstract.** Long-term observations of spring discharge provide an alternative to estimate the evolution of groundwater resources based on observational data at catchment scale. Karst springs can be found in large parts of Europe covering all
climate zones of the mid latitudes. Continuous spring discharge measurements are holistic signals, representing both fast and slow flow components, typical of karstic environments. Due to relatively short response times, karst systems are focus points for enhancing the understanding of the impact of climate change on groundwater resources. This work analyses observational data (precipitation, temperature and discharge) of more than 50 springs spatially distributed over Europe to give a continental overview of changes in groundwater resources in karst areas. The work focuses on two different periods of 20 and 40 years,
to identify any possible acceleration or moderation in changes. For both periods a trend analysis of the observational data, using Mann-Kendall and Sen's slope, was performed on the entire time series and per season. Possible process changes were considered by analysing also trends in high and low flow values. Structural differences of the systems were considered by using two indices related to the storage and inertia of the system. In combination, these indices were able to i) highlight structural differences and ii) characterize karst systems accordingly. The results show that the sensitivity of karst aquifers to
climate change is not controlled by their degree of karstification. Long-term trends in spring discharge calculated in this study follow the general pattern of river discharge found in literature, but the last 20 years deviate from this behaviour. During this period, increasing temperature plays a more important role in the evolution of spring discharge than changes in precipitation. These results are discussed in relation to the indirect influence of other drivers such as changes in land use or land cover, specific regional conditions but also changes in processes related to groundwater recharge and storage, providing
insights for assessing groundwater recharge in the past and in the future.

## 1 Introduction

The impact of global change on fresh water availability and quality is mostly associated with changes in hydrological extremes, for example droughts or floods (e.g. Blöschl et al., 2019; Vicente-Serrano et al., 2019; Tramblay et al., 2020; Lorenzo-Lacruz et al., 2022), whereas the future evolution of groundwater recharge - as a key process for sustainable water supply by
groundwater - is currently highly uncertain (IPCC, 2021). The complexity and manifold of processes related to groundwater



recharge, make it highly variable in space and time with interconnections to various drivers (see Moeck et al., 2020; Riedel and Weber, 2020; Barthel et al., 2021). In addition, direct measurements of spatial distributed actual recharge are not possible. This makes it difficult to identify trends on regional scale even by analyzing a comprehensive global data set of groundwater recharge measurements (Moeck et al., 2020).

Hydroclimatic conditions, e.g. precipitation and temperature, are usually considered as main drivers of land surface water fluxes and make groundwater recharge highly vulnerable to climate change (Mohan et al., 2018). However, over the last decades, human impact on groundwater recharge increased due to changes of water consumption and land use (e.g. Haas and Birk, 2019). The latter is closely connected to net increases in evaporation. In large parts of Europe, impacts of anthropogenic activity are comparable to those caused by changing hydroclimatic conditions (Teuling et al., 2019), reaching the point where

they prevail over changes in natural land surface fluxes (Riedel and Weber, 2020). Approaches for groundwater recharge modelling allow for the consideration of these changing conditions on a larger scale (e.g. Lanini and Caballero, 2021; Martinsen et al., 2022; Seidenfaden et al., 2023). However, they have the disadvantage that a validation of derived values with in-situ observations is generally not possible. This leaves hydrological time series based on observations as an essential reference point for studying changes in water resources.

Long-term observations of spring discharge provide an alternative to estimate the evolution of groundwater resources based on observational data at catchment scale. In this context, karst aquifers appear to be of special interest due to their specific properties, closely related to the soluble (carbonate) rocks, generating a hierarchical organised groundwater drainage network over large areas (e.g. Palmer, 1991, Ford and Williams, 2007). Therefore, continuous measurements of karst spring discharge give a holistic output signal of the aquifer system combining all the processes transforming the recharge signal on the way

through the system to the outlet. Different to most other types of aquifer systems, groundwater recharge in karst areas is thus highly influenced by preferential flow in fissures and open conduits and needs to be divided into concentrated and diffuse processes (e.g. Hobbs and Smart, 1986). Concentrated infiltration into the system generally leads to fast flow dynamics which makes it potentially possible to detect the impact of changes in groundwater recharge on water resources. Following this reasoning spring discharge can be used as a robust regional climate indicator (Fiorillo et al., 2021) and due to their sensitivity,

karst aquifer can be sentinels of global changes (Binet et al., 2020; Binet et al., 2022).

In Europe, karst outcrops cover an absolute area of 2.17 million km$^2$ accounting for a total area share of 21.8 % (Goldscheider et al., 2020). The contribution of water from karst areas to the national fresh water supply varies widely between 5% and 50% (Hartmann et al., 2014), making karst aquifers together with alluvial formations to the most efficient source of fresh water (Bakalowicz, 2005). For water management purposes but also securing ecological flow downstream, low flow characteristics

in karst systems and their development are of special importance. Less in focus are high flow characteristics even though flash flood in karst systems can cause hazards for human life (Maréchal et al., 2008).

Most of the global karst springs with long observation records are located in Europe (Olarinoye et al., 2020). Spring discharge analysis is a common tool to characterize karst systems but can also be used to analyse changes in hydroclimatic conditions or calculated recharge values. Over a long period, discharge of European karst springs seems to decline caused by rising





temperatures and the consequent reduction of snow contribution (Lorenzi et al., 2022, Fan et al., 2023) and increased evapotranspiration (Leone et al., 2021) rather than changes in precipitation pattern. The latter only seems to have an impact on drought frequency (Leone et al., 2021) and peak discharge (Fan et al., 2023). Other studies from European karst areas highlight the influence of land use or land cover changes, e.g. large-scale forest disturbance (Kovacic et al., 2020; Vilhar et al., 2022), changes in intensification of agricultural use (Palacios-Cabrera et al., 2022), and water abstraction (Charlier et al., 2015). These

influences have the ability to mask the long-term influence of climate change on spring discharge. Furthermore, several studies on Italian karst springs highlighted a strong correlation between spring discharge and large-scale atmospheric circulations, e.g. North Atlantic Oscillation (NAO). Negative (winter) NAO values have been resulting in an increase in spring discharge since roughly 2008 correlated to (De Vita et al., 2012; Fiorillo and Guadango 2012; Fiorillo et al., 2015; Fiorillo et al., 2021). However, different large-scale atmospheric circulations influence the discharge from Italian karst springs resulting in a

complex periodicity ranging from two years to multi-decadal cycles (De Vita et al., 2012). According to the authors, the impact on spring discharge is high, accounting for a variability of roughly 30%. Differences in the response of the karst systems are detected, which are generally explained by variations in storativity or inertia of the karst system (Fiorillo and Guadango 2012; Fiorillo et al., 2021; Lorenzi et al., 2022).

Changing hydroclimatic conditions are thus projected to jeopardize fresh water resources in most karst areas in Europe by

reducing absolute groundwater recharge until the end of this century (Hartmann et al., 2017). Although the long-term impact on groundwater resources in other systems is still highly uncertain, it is most likely that climate change will affects karst water resources and therefore the reliable water supply to millions of people negatively. In this context, this paper presents a continental overview of changes in groundwater resources in karst areas over the past decades, based on more than 50 springs spatially distributed over Europe. This study is a multi-decanal trend analysis of hydroclimatic observational data for European

karst systems and designed to answer the following research questions:

- Does discharge from European karst areas change uniformly over time or is it possible to detect regional patterns?
- Do significant changes in overall spring discharge have an impact on low and high flow conditions?
- Is it possible to identify sets of karst storage properties which are particularly sensitive to climate change?

**2 Data and methods**

Karst outcrops drained by major karst springs can be found in large parts of Europe (Fig.1; Chen et al., 2017) covering all climate zones of the mid latitudes. Hydroclimatic conditions together with the geological history play an important role in the evolution of karst aquifers leading to a complex pattern of systems on a continental scale. To evaluate the impact of changing climatic conditions of different types of karst systems, this study focuses on publically available spring discharge data from several European countries. To be able to analyse possible impacts of climate change on karst groundwater resources during

the last decades, the following method was applied. Long-term trends in karst spring discharge were assessed for all European springs with time series fulfilling the requirements indicated in section 2.1. The assessment was performed over two different





time periods – described herein – following the maxim of a) having as many discharge records as possible among the available data and b) comparing results between two different periods in order to identify any possible acceleration or moderation of changes. Changes were analysed using the Mann-Kendall test and Sen's slope computation (Section 2.3) for monthly spring

discharge and annual extremes, expressed by different quantiles. The two investigated periods are, first a period of 40 years (starting at 01.01.1982) to cover long-term trends and second a shorter period of 20 years (starting date: 01.01.2002). The shorter period allows for an investigation of changes on the same time scale that those described for Italian karst springs (e.g. Fiorillo et al., 2021). Further, trends were computed on monthly precipitation and temperature to identify changes in the input signal of the systems. To explore the impact of groundwater dynamics on observed trends, two specific indices related to

storativity and inertia (Section 2.1.1/2.1.2) were calculated for each period.

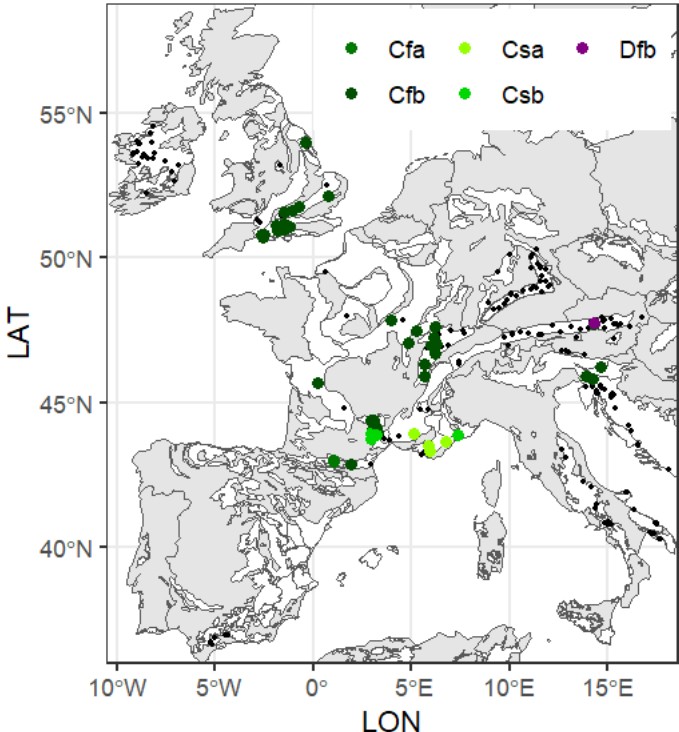

**Fig 1: Karst spring location (black dots) according to the WOKAS database (Olarinoye et al., 2020) and carbonate rock outcrop (white area) in Europe according to WoKaM (Chen et al., 2017). Springs used in this analysis are coloured. Colours indicate the climate zone according to the Koepper-Geiger classification (1986-2010) in Table 1.**

**2.1 Karst spring discharge**

Spring discharge data were derived from the World Karst Spring Hydrograph dataset (WOKAS; Olarinoye et al., 2020) and the French national database Hydroportail (https://www.hydro.eaufrance.fr/). The WOKAS dataset combines spring discharge data from national agencies as well as time series collected by many research groups around the world. In both databases, spring discharge is considered in its broadest sense which means that measurements might also be taken further downstream




of the spring. From both datasets, all time series satisfying the minimum requirements for the two periods of 20 years (starting

at 01.01.2002) and 40 years (starting at 01.01.1982) were selected. The minimum requirements were a) continuous daily

measurements with b) less than 10% of missing data over the 20- respectively 40-year period. No further pre-processing, for

example the interpolation of missing data, was done prior the analysis. In total 54 springs from 4 different countries and 22

springs from 4 different countries fit the requirements for the short, respectively the long period (Fig. 1). Spring names and

basic information regarding the time series and location of the spring can be found in Table 1. Note that 9 time series from

French springs start during the first or the second year, all other time series were unified to start at the 1st of January and end

20 respectively 40 years later, on the 31st of December 2021. For Austrian springs, data is only available until the end of 2019.

Most springs are located in the temperate oceanic climate zone (Cfb) covering large parts of Western Europe. Springs located

in different climate zones can be found in the Pyrenees (Cfa – warm temperate climate), parts of Austria (Dfb – humid

continental climate) and in Southern France (Csa/Csb- Mediterranean climate). Some of the investigated karst areas, mainly

in Austria and the Grands Causses region in Southern France, are located along climate zone boundaries.

**Table 1: Overview of the European karst springs used in this analysis. Climate zones are according to the Koeppen-Geiger classification (1986-2010). Mean discharge is calculated from the daily values either on the 40-year period or for shorter time series**
**on the 20-year period.**

| Country | Name | Lat | Lon | Climate zone | $Q_{mean}$ (m³/s) | 40y | 20y |
|---------|------|-----|-----|--------------|-------------------|-----|-----|
| AT | Piessling Ursprung | 47.69 | 14.28 | Cfb | 2.15 | X | 0 |
| AT | Rettenbachquelle | 47.76 | 14.31 | Dfb | 1.06 | 0 | X |
| FR | Aiguebelle | 43.93 | 3.06 | Csb | 0.12 | 0 | X |
| FR | Aliou | 42.99 | 1.05 | Cfa | 0.43 | X | 0 |
| FR | Arcier | 47.27 | 6.12 | Cfb | 1.14 | 0 | X |
| FR | Argens | 43.50 | 5.91 | Csa | 0.38 | X | X |
| FR | Baget | 42.96 | 1.03 | Cfa | 0.45 | X | 0 |
| FR | Balastière | 43.97 | 3.03 | Csb | 0.07 | 0 | X |
| FR | Barbade | 44.18 | 3.09 | Cfb | 0.08 | 0 | X |
| FR | Bastide | 44.30 | 3.07 | Cfb | 0.03 | 0 | X |
| FR | Bèze | 47.47 | 5.27 | Cfb | 3.81 | X | X |
| FR | Bleue Dortan | 46.31 | 5.66 | Cfb | 1.21 | 0 | X |
| FR | Boundoulaou | 44.07 | 3.05 | Cfb | 0.23 | 0 | X |
| FR | Cainea | 43.88 | 7.35 | Csb | 0.03 | 0 | X |
| FR | Caramy | 43.35 | 5.92 | Csa | 0.06 | 0 | X |
| FR | Ceras | 43.75 | 2.96 | Csb | 0.99 | 0 | X |
| FR | Cernon | 43.96 | 3.15 | Cfb | 0.19 | 0 | X |



| | | | | | | | |
|---|---|---|---|---|---|---|---|
| **FR** | Doubs | 46.71 | 6.21 | Cfb | 1.89 | X | X |
| **FR** | Dragonnière | 43.95 | 2.94 | Csb | 0.08 | 0 | X |
| **FR** | Duc | 44.40 | 3.08 | Cfb | 0.04 | 0 | X |
| **FR** | Durzon | 43.99 | 3.26 | Cfb | 1.52 | 0 | X |
| **FR** | Esperelle | 44.12 | 3.21 | Cfb | 1.01 | 0 | X |
| **FR** | Font de Champdamoy | 47.61 | 6.19 | Cfb | 2.33 | 0 | X |
| **FR** | Fontaine de Vaucluse | 43.92 | 5.13 | Csa | 15.51 | 0 | X |
| **FR** | Fontestorbes | 44.89 | 1.93 | Cfb | 1.88 | X | X |
| **FR** | Fosse Dionne | 47.86 | 3.97 | Cfb | 0.30 | 0 | X |
| **FR** | Fousette | 43.91 | 3.12 | Csb | 0.17 | 0 | X |
| **FR** | Gloriette | 43.91 | 3.18 | Csb | 0.09 | 0 | X |
| **FR** | Groin | 45.89 | 5.69 | Cfb | 3.15 | X | X |
| **FR** | Ladoix | 47.07 | 4.88 | Cfb | 0.22 | 0 | X |
| **FR** | Ladoux | 44.08 | 3.06 | Cfb | 0.27 | 0 | X |
| **FR** | Lestang | 44.41 | 3.02 | Cfb | 0.21 | 0 | X |
| **FR** | Lison | 46.96 | 6.01 | Cfb | 5.07 | X | X |
| **FR** | Loue | 47.01 | 6.30 | Cfb | 9.56 | 0 | X |
| **FR** | Mayrinhac | 44.39 | 2.95 | Cfb | 0.13 | 0 | X |
| **FR** | Mouline | 44.99 | 1.05 | Cfb | 0.50 | 0 | X |
| **FR** | Segala | 44.36 | 3.02 | Cfb | 0.17 | 0 | X |
| **FR** | Sorgue | 43.88 | 3.19 | Csb | 1.03 | 0 | X |
| **FR** | Touvre | 45.66 | 0.26 | Cfb | 13.05 | X | X |
| **FR** | Tuves | 43.64 | 6.79 | Csa | 0.06 | 0 | X |
| **FR** | Verneau | 46.98 | 6.00 | Cfb | 0.44 | X | X |
| **GB** | Avon | 50.93 | -1.07 | Cfb | 15.75 | X | X |
| **GB** | Brett | 52.14 | 0.79 | Cfb | 0.13 | X | X |
| **GB** | Cheriton | 51.09 | -1.18 | Cfb | 0.69 | X | X |
| **GB** | Ewelme | 51.62 | -1.07 | Cfb | 0.04 | X | X |
| **GB** | Hooke | 50.80 | -2.66 | Cfb | 0.02 | 0 | X |
| **GB** | Manor Farm Brook | 51.57 | -1.45 | Cfb | 0.01 | 0 | X |
| **GB** | S. Winterbourne | 50.71 | -2.53 | Cfb | 0.12 | 0 | X |
| **GB** | Sydling Water | 50.8 | -2.52 | Cfb | 0.20 | X | X |
| **GB** | Test Broadlands | 50.97 | -1.50 | Cfb | 10.88 | X | X |





| | | | | | | |
|---|---|---|---|---|---|---|
| **GB** | Test Chilbolton | 51.15 | -1.45 | Cfb | 5.59 | 0 | X |
| **GB** | Wendover Spring | 51.77 | -0.74 | Cfb | 0.09 | 0 | X |
| **GB** | West Beck | 53.99 | -0.37 | Cfb | 2.38 | 0 | X |
| **GB** | Wylye | 51.10 | -1.88 | Cfb | 4.00 | X | X |
| **SI** | Hubelj-Ajdovščina I | 45.90 | 13.91 | Cfb | 2.80 | X | X |
| **SI** | K. Bistrica-Kamnik | 46.22 | 14.62 | Cfb | 7.04 | X | 0 |
| **SI** | Malenščica-Malni | 45.82 | 14.25 | Cfb | 6.15 | X | X |
| **SI** | Unica-Hasberg | 45.83 | 14.26 | Cfb | 20.89 | X | X |

Monthly mean discharge was calculated for each spring (Table 1) and used for the trend analysis. Further, trends were calculated for the annual minimum and maximum flow. These values were derived by accumulating the discharge below the 10th quantile (Q10) respectively above the 90th quantile (Q90) using the R package *stats*. In addition, the karst systems were classified based on two indices derived from the daily time series. In general, karst systems are classified based on the description of flow patterns, e.g. Mangin (1975) or Quinlan and Ewers (1985), which are a combination of dominant recharge processes, storativity and flow of different karst system compartments (Hoobs and Smart, 1986). All these factors highly depend on the hydraulic properties closely related to the maturity or degree of karstification of the system. To classify the springs based on dominating discharge component and storativity, two frequently used indices were computed.

Firstly, to describe the storage capacity of karst systems, daily discharge was filtered to separate quick and slow flow components and to compute the baseflow index (BFI). Traditionally the slow component is interpreted as baseflow (Smakhtin, 2001), in karst hydrology often conceptually described as the outflow of a single reservoir as a function of the active storage volume (Maillet, 1905). Many different separation methods exist (e.g. Sloto and Crouse, 1996; Rutledge, 1998; Piggott et al., 2005; Eckhardt, 2005). Following the proposal of Ladson et al. (2013), here the "standard approach" by Lyne and Hollick (1979) was chosen, using the R package *BFI* (Ladson et al., 2013). This method uses a one-parameter recursive digital filter. The baseflow volume and therefore BFI is sensitive to the filter coefficient α which ranges between 0.9 and 0.98. Here, α was set to 0.925 for all springs since the analysis focus only on the evolution of the baseflow and not a quantitative comparison between different springs or periods. In case of larger gaps, the values are calculated individually for the segments and the BFI for the entire time series is the weighted average of the segments.

Secondly, to describe the inertia of karst systems, the memory effect of the karst systems was calculated. It is determined by using a threshold of 0.2 in the autocorrelation function below which the signal is considered as undistinguishable from noise (Mangin, 1984). The memory effect has been widely used to characterize the storativity of karst systems and to compare response time between different systems (e.g. Larocque et al., 1998; Padilla and Pulido-Bosch, 1995; Fiorillo and Doglioni, 2010; Dubois et al., 2020; Cinkus et al., 2021; Bailly-Comte et al., 2023). One autocorrelation function was calculated for each period and spring using the R package *stats*.



## 2.2 Precipitation and temperature

Precipitation and temperature data were derived from the daily gridded observational dataset for Europe (E-OBS; Cornes et al 2018). The E-OBS dataset provides data with a spatial resolution of 0.1-degree regular grid based on the interpolation of hydroclimatic variables from a network of European meteorological stations. Currently, the dataset covers the period between 1950 and 2022. For every karst spring, the climate variables were selected based on the coordinates of the spring. This means that in areas with high spring density, one grid cell can represent hydroclimatic variables for more than one spring. Daily values of precipitation and temperature were accumulated respective averaged to obtain monthly and seasonal values. Even though the study covers entire Europe and therefore different climate zones, standardized seasons (winter: DJF, spring: MAM, summer: JJA, and autumn: SON) were used for comparison purposes.

## 2.3 Trend analysis

The Mann-Kendall test (Kendall 1948; Mann, 1945) is frequently used for the trend analysis of observational data and climatic indices. This includes the observational E-OBS dataset, which has been analysed for different timely resolution for example monthly, seasonal, and yearly before (e.g. Peña-Angulo et al., 2020). However, to compare the observational data at the location of the springs over the two defined periods, the Mann-Kendall test was calculated for spring discharge and hydroclimatic variables. The ranked-based non-parametric Mann-Kendall test determines the significance of monotonic trends by comparing the difference between measurements earlier and later in time. As a non-parametric test, it can be used for time series with non-normal distribution and is therefore often applied in hydrological studies. Here, the modified Mann-Kendall test after Hamed and Rao (1998) was used, which includes a variance correction approach to account for autocorrelation in the time series. To overcome the sensitivity of the confidence level of the Mann-Kendall test, two different statistical significance level were set, the first one with p-values of 5 %, and the second one with p-values of 10 %. In addition, Sen's slope (Sen, 1968) is used to calculate the sign and slope of the calculated trend. Mann-Kendall test and Sen's slope were used because they are frequently applied for monotonic trend analysis in both climatic and hydrological studies. They have the advantage of being independent of the data distribution, and hence applicable to non-normal distribution as found for monthly discharge data from karst springs (e.g. Fiorillo et al., 2021). Both statistical tests are included in the R package *modifiedmk* (Hamed and Rao, 1998).

## 3 Results

### 3.1 Trends in monthly spring discharge and climate variables

Only six of the analysed springs exhibit significant changes in discharge over the last 40 years. The three negative changes have a higher significance (at the 5% probability level) than the two positive ones (at the 10% probability level). Apart from two springs, all springs with discharge records of 40 years are located in the temperate climate zone (Cfb). Both springs, one



located in the Mediterranean climate zone (Csa), the other one in the humid subtropical climate zone (Cfa), have decreasing spring discharge. The latter one (spring Fontestorbes) is one of the springs draining a mountainous catchment in the Pyrenees. The other two springs with decreasing spring discharge (spring Lison and Verneau) also drain karst systems in a mountainous region, the French Jura mountains. The only other spring located in a mountainous region is the spring Pissling Ursprung,
draining a karst system on the northern side of the Austrian Alps. This spring is one of two springs with increasing discharge trends. The other one is in England (spring Cheriton).

Discharge trends of the last 20 years (Fig. 2b) differ locally but also on a continental scale from the long-term trends. Positive trends are dominant in large parts of Europe. Out of 15 springs with positive trend, one each is in the Csa and Csb climate zone, the rest are in the Cfb climate zone. Most of them are karst springs in the Grands Causses region, a high plateau under
Mediterranean influence in southern France. Apart from there, positive trends can also be found in other parts of France and in England. Three springs have a negative sign in discharge, spread out over Europe. In addition, none of the springs with trends over the 40-year period has any significant changes in the 20-year period.







**Figure 2: Trends in discharge (a,b), precipitation (c,d) and temperature (e,f) over the 40-year (1982-2021 – left) respectively the 20-year (2002-2021 – right) period. Red symbols indicate negative trends while green symbols indicate positive trends.**





Despite the number of springs with significant trends in discharge, precipitation is rather stable over both periods. As presented
in Fig 2c) and d), precipitation changes only locally. Over the period of 40 years, only 6 karst areas, represented by EOBS grid
cells, have a significant precipitation trend. Three locations can be found in southern England in the same area of the spring
Chariton. However, the exact location of the Chariton station does not show a significant increase. The only region with
negative precipitation trends is the French Jura mountains, where two of the springs with negative discharge are located. Over
the last 20 years, the negative precipitation trend in the Jura Mountains is slowed down and no significant changes can be
detected in this area. Further south in the Grands Causses region, precipitation decrease in the far north of the region, however
only at the lower significance level. The number of positive trends in grid cells located in the UK also decreases to one with
lower significance level.

Air temperature shows a uniform significant increasing trend in Europe over the 40-year period (Fig.2e). During the last 20
years (Fig. 2f), temperature increases at most of the spring locations, apart from England where no trends are observed.

**3.2 Seasonal changes in monthly spring discharge and climate variables**

Spring discharge varies widely between European karst spring (see Tab. 1). To account for the differences in mean discharge,
a normalized Sen´s slope (seasonal Sen´s slope/$Q_{mean}$) for discharge was used to analyse changes in seasonal discharge on the
continental scale. In Figure 3, top-right and bottom-left corners of each seasonal subplots indicate similar positive and negative
Sen´s slopes for normalized seasonal Q and seasonal P, respectively. For simplification reasons all locations are summarized
in the same plot regardless the climatic zone they belong to. Following the results of Fig. 2, air temperature was not included
in the seasonal analysis, assuming that its global increase for all spring was not a key explaining factor for inter-annual
variability of discharge.

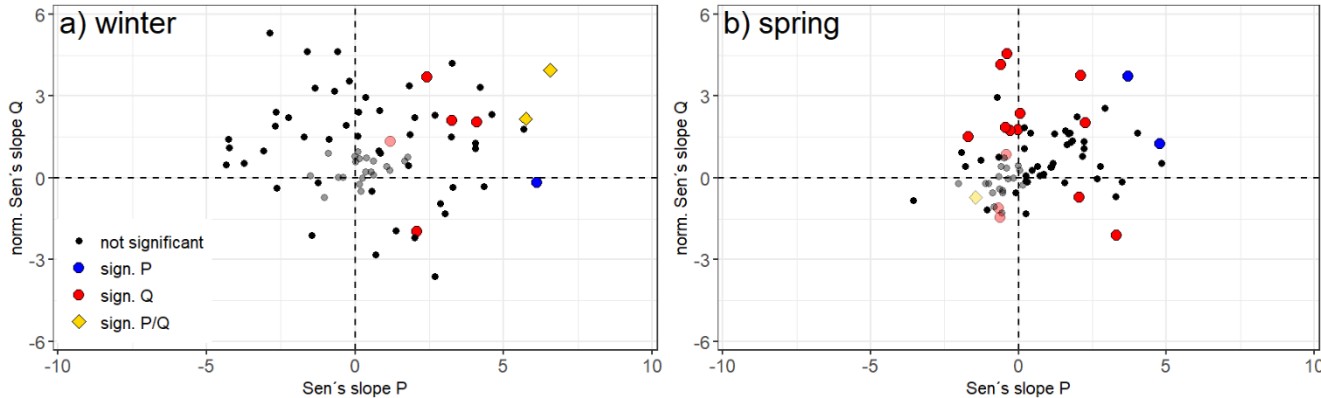





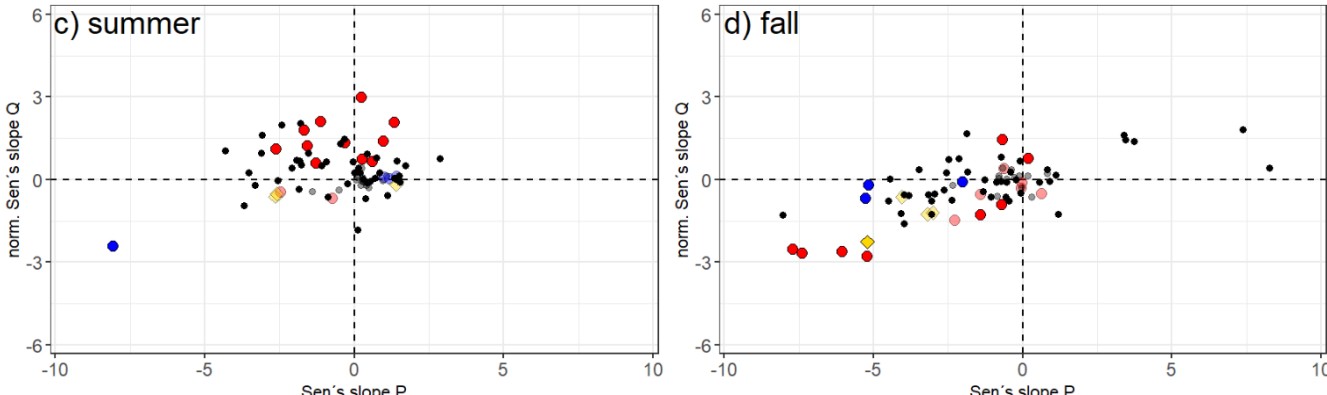

**Figure 3: Comparison of changes in precipitation and discharge trends for a) winter, b) spring, c) summer, and d) fall. Springs indicated by red dots have significant changes in discharge, by blue dots in precipitations and diamonds have significant changes in both seasonal precipitation and discharge. Bold symbols represent the 20-year period.**

Overall, significant changes in both P and Q at the same time are rather few and occur only at a small percentage of springs. With one exception (in summer over the 40-year period) the signs of these significant trends in precipitation and discharge are similar. Overall, significant changes in Q are predominant but the sign shows a high variation among springs and seasons. Winter is the season with the least significant changes over both periods and the highest variation in Sen´s slopes for both discharge and precipitation trends. However, most springs have positive Sen´s slopes and therefore an overall increasing trend in discharge during winter. Sen´s slopes of precipitation changes range widely between negative and positive values. In terms of sign and overall trends, there are no major differences between the two periods. However, the magnitude of changes is highly elevated during the short period, a pattern which can be found in all seasons except of fall, for which variations in the long period are high as well. However, at least in winter most changes are not significant. Spring and summer, on the other hand, show a clear increase of significant discharge changes during the last 20 years. For both seasons, most of these changes are significantly positive. Clear trends in precipitation are missing, even though spring seems to become slightly wetter and summer drier on a continental scale. In both season, differences between the springs are minor compared to the winter. The clearest trend of all seasons is the precipitation decrease in fall, often combined with a dominant decrease in discharge. These trends are equally developed for both periods, slightly amplified for the last 20 years. Therefore, it can be stated that the fall is the only season with a clear long-term connection between precipitation and discharge, with a high sensitivity of spring discharge changes to changes in precipitation.

**3.3 Low and high flow conditions**

To analyse changes in low and high extremes, a trend analysis on the 0.1 and 0.9 percentile of spring discharge is done. In Figure 4, trend changes are expressed by Sen's slopes in $m^3y^{-1}$, with changes in high (Q0.9) and low (Q0.1) flow conditions on the y-axis and x-axis, respectively. Relative changes in annual high and low flow discharge and annual discharge are difficult to compare since the spring differ in discharge dynamics influencing mean annual discharge (cf. Table 1). Therefore,



again normalized values are used in Fig.4. The figure consists of four different quadrants where the upper right and lower left represent a unison shift of discharge to a higher or lower stage, respectively. The other two quadrants represent conditions where the fluctuation in discharge either increases (upper left) or decreases (lower right).

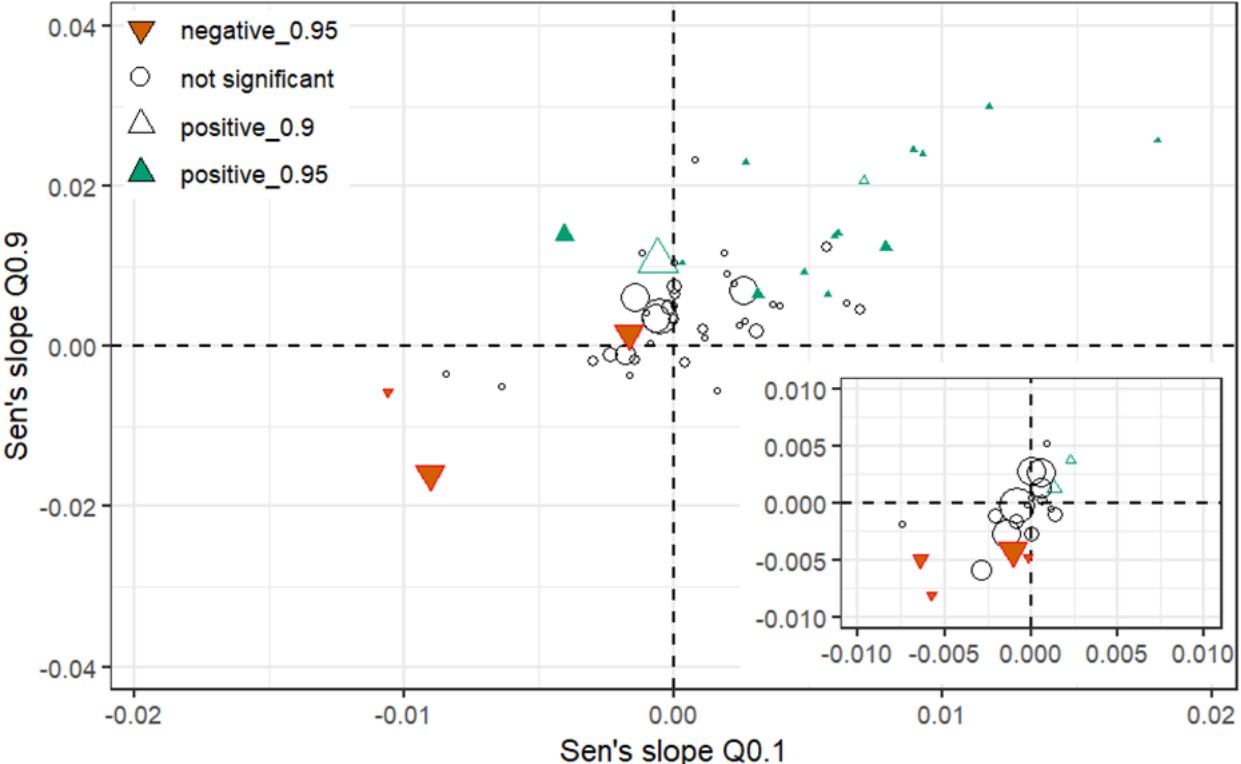

**Figure 4: Distribution of high flow (Q0.9) and low flow (Q0.1) trends for a) the 20year period and b) the 40year period. Shape and**
255 **colour of the symbols refer to the monthly discharge trends presented in Figure 2a and b. The size of the symbols represents the size**
**of the spring expressed by $Q_{mean}$.**

Comparing the trend development of extremes between the two different periods gives valuable information on recharge patterns. For springs with significant changes in annual spring discharge over the last 40 years, the entire spring discharge shifts unison either to a lower or higher stage depending on sign of the global trend. All springs with negative trends in annual

discharge are located in the bottom left, which is associated with discharge decreases in both low and high flow. Also most other springs without significant trend follow this pattern and can be found either in the top right or bottom left. Only two springs differ from this general pattern, showing a slightly negative Sen's slope for high flow discharge in connection to positive low flow discharge. For the 20-year period, several springs deviate from this distribution including three with significant changes in annual spring discharge. All these springs are located in the top left quadrant with opposing trends in

low and high flow discharge. In two cases the increase in high flow discharge leads to a significant increase in annual spring discharge, whereas one spring even shows significant decreasing spring discharge despite the increase in high flow discharge. All other springs located in the lower left or upper right quadrant are following the pattern observed for the long period. Springs



with significant increasing trend and decreasing can be solely found in the upper right quadrant and lower left quadrant, respectively.

**3.4 Sensitivity of different karst systems to changing climatic conditions**

Due to large differences in hydraulic properties, karst systems respond differently to input signals. To characterize the springs according to their hydraulic properties, a simplified characterization based on BFI and memory is presented in Figure 5. Each spring is represented by one point in the coordinate system. Each axis represents one index starting with zero in the origin and increases the values along the axis. The resulting coordinate system allows for the simplified classification of the systems with two endmembers in the lower left and the top right corner of the system. The lower left corner represents system with a low fraction of the slow flow component and low storativity and therefore mature karst systems, whereas the top right corner represents fissured system with a high storage and a high degree of diffuse recharge.

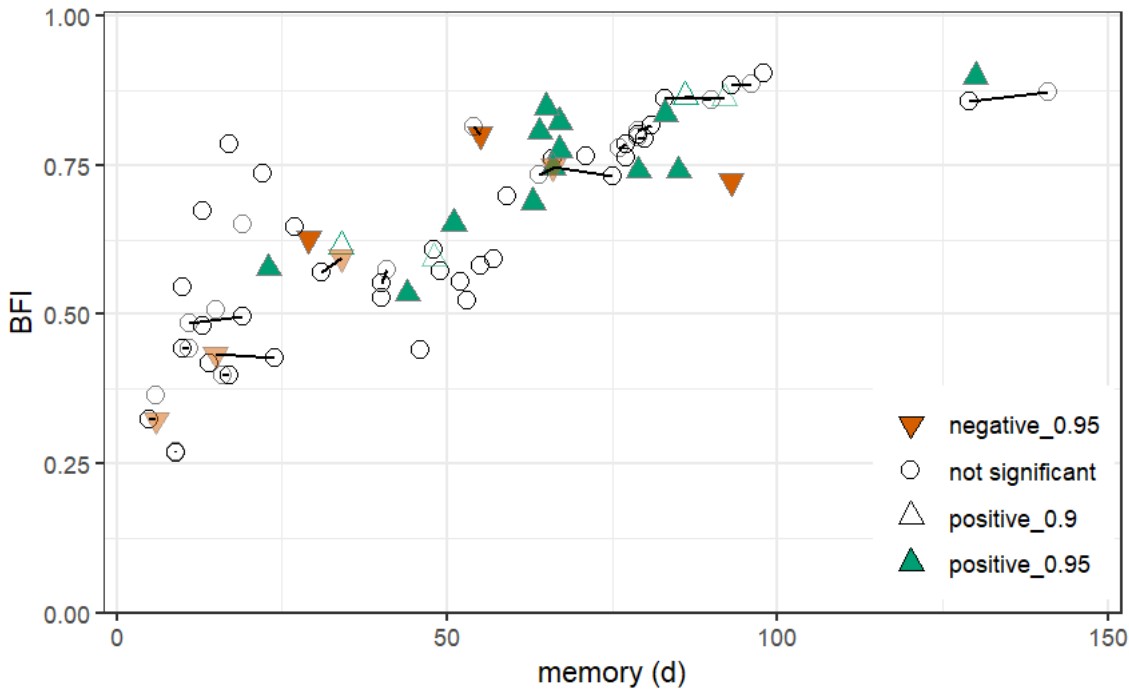

**Figure 5: Classification of karst systems based on base flow index (BFI) and memory value. Bold symbols represent the 20-year period, points connected by a line are the same spring at the two different periods.**

Figure 5 captures the variety of European karst systems ranging from fast responding systems with memory values of only a few days with a low fraction of slow flow components to fractured systems characterized by a high fraction of low flow components and high inertia. All springs follow one general trend with some alternations at the end of the spectrum. This general trend is a vertical line from the lower left to the top right. Among the fissured systems, in the range of BFI > 0.85, two spring alternate from the general trend by higher memory values. A second alternation can be seen among the mature karst





systems, where several springs show increased fraction of slow flow components. Despite the obvious differences in hydraulic properties, significantly positive and negative trends cover the entire spectrum of springs. Only springs with extremely high values of both BFI (> 0.8) and memory (> 90 d) tend to increasing or stable discharge. Based on these results of a continental discharge dataset, it is not possible to link hydraulic properties to climate resilience.

## 4 Discussion

Summarizing the continental trends in hydroclimatic variables, it can be stated that trends in air temperature are the ones that are most pronounced in both analysed periods. Over the last 40 years, air temperature basically increases in all recharge areas of European karst springs. During the last 20 years this trend is somehow slowed down in some areas, especially England. Significant trends in precipitation are rare for both periods. Over the 40-year period, significant trends can be detected only locally, and the significance and spatial occurrence further decreases over the last 20 years.

Considering only general trends in observational data over longer periods makes it difficult to explain changes in discharge from karst areas by hydroclimatic changes only. This might be due to the fact that additional factors such as groundwater abstraction and other anthropogenic interferences like land use changes or changes in agricultural crop rotations in karst areas are not considered but also due to the fact that trends in spring discharge might not only be influenced by long-term changes of hydroclimatic conditions but also short-term (e.g. seasonal) changes in processes related to groundwater recharge and storage. Following a general comparative analysis with river discharge trends, possible process changes will hence be discussed in a regional context.

### 4.1 Comparative analysis with surface water

Due to the relative high abundance of hydrometric stations, plenty of trend analysis of river discharge have been done on continental and local scale. These studies can be used as comparison for the trends in karst areas. In fact, river discharge trends on a European scale show a clear regional pattern closely connected to climatic drivers over the last few decades. The increase in (extreme) precipitation results in positive trends in river discharge at the majority of hydrometric stations in north-western Europe (Harrigan et al., 2018; Blöschl et al., 2019) even though Central and Southern England as well as Northern France only show a few significant trends (Vicente-Serrano et al., 2019). In southern Europe, regional river discharge trends are spatially negative as a consequence of increasing temperatures and large decline in precipitation (Blöschl et al., 2019; Vicente-Serrano et al., 2019). One exception are the Pyrenees Mountains, where – on the contrary to karst springs - streamflow trends are largely not significant for the period between 1980 and 2013 based on 67 river gauging stations covering the north as well as the south side as concluded by Clavera-Gispert et al. (2023). Even though the spatial coverage of karst spring data over a long period is comparably low, the results presented in Fig. 2 showed that it is possible to detect the long-term impact of changing hydroclimatic conditions on water resources in karst areas on a continental scale. The general pattern presented for spring





discharge in this study are in line with trends with another component of the terrestrial hydrological cycle, namely surface runoff.

High flow conditions in the karst system, described by the 10th and 90th percentile, follow the general trend of monthly discharge over the long period (Fig. 4). This is again coherent with results of continental studies on European river systems.

The significance in flood trends show a distinct regional pattern, which is somehow equal to the trends in mean river discharge. Increasing trends in flood magnitude and frequency are dominant around the Atlantic (Mangini et al., 2018), with positive trends in high flow indices for all seasons except spring between 1985 and 2014 in the UK (Harrigan et al., 2018). Following this pattern, river discharge in south-east France has mainly negative trends (Mangini et al., 2018). Furthermore, decreasing flood magnitudes are discovered in the southern part of the Alps (Mangini et al., 2018). In the Mediterranean, flood frequency

is decreasing but the magnitude shows an increasing trend, both for moderate (95th percentile) and high (99th percentile) floods (Mangini et al., 2018; Tramblay et al., 2019). Changes can also be detected in the characteristics of floods which is most prominent in a considerable increase of medium to large flash flood occurrence in Europe in the 21st century compared to the 1980s (Owen et al., 2018) even though major floods (25 to 100-year return period) show an overall but insignificant increase (Hodgkins et al., 2017).

The consistency in European flood trends and karstic flows is a sign that groundwater recharge in karst areas follows the same trend as surface runoff on a catchment scale. A clear sign that global changes in hydroclimatic variables impact both runoff and infiltration components of the water cycle. But we can assume that the concentrated infiltration typical of karstic zones may reinforce this pattern, and that this would probably be buffered for other types of more inertial aquifers.

However, the evolution of trends between both periods in high and low flow in karst areas indicate process changes in the last

decades (Fig.4). Over the 40-year period most of the springs follow roughly a straight line from the lower left to upper right: Along this line, springs with overall decreasing discharge trend are located in the lower left, and the once with increasing discharge in the upper right. This indicates that the entire time series transverses to a higher or lower level without changes of the fluctuation pattern. In opposite to this, more springs deviate from this behaviour during the last 20 years. All these springs have a positive high flow and negative low flow trends, which means that annual fluctuations between high and low flow

conditions increase. Some of the springs even have significant trends in overall discharge which can be explained by an increase in high flow events for overall increasing trends and lower baseflow but more extreme events for overall decreasing trends. This is a strong indication for changes in the concentrated and diffuse recharge partitioning and therefore an indication of changes in precipitation patterns. However, considering the results from the analysis of the indices closely related to the maturity or degree of karstification (memory effect and BFI; Fig. 5), it becomes obvious that not only systems with fast flow

component are impacted. Hence, the sensitivity of karst aquifers to climate change is not controlled by their degree of karstification and the effects linked to hydraulic properties seem to be masked by the regional effect of hydroclimatic changes on different time scales.





## 4.2 Hydroclimate-induced changes in karst water resources

Changes on a regional scale depend on the climate zone but also specific regional conditions. Even though a thorough analysis

on a local scale is not focus of the analysis, evidence for some of the described discharge trends can be given. According to the results, fall is the seasons of highest sensitivity of spring discharge changes to changes of precipitation. (Fig. 3). A likely explanation would be the combination of two phenomenon leading to severe consequences: i) the depletion of the aquifers in fall due to the warmer spring/summer temperatures, and ii) a lower recharge at the start of the hydrological cycle. This can be discussed at a smaller scale, for regions specific areas with consistent spatial trends.

Over the last 40 years, discharge decreased in the springs Lison and Vernean, two springs with long time series (40 years) located in the same part of the Jura Mountain. Looking at the general trends in the area (Fig. 6), temperature and therefore evapotranspiration increased in all seasons over the last 40 years, resulting together with significantly decreasing fall precipitation for the entire region in lower discharge in fall. In the entire region, the distribution of precipitation changes to more precipitation in winter and spring and less during the other two seasons. Over 40 years, the partly significant decrease in

fall precipitation might explain the local precipitation decreases in the area (Fig. 2d). During the last 20 years, precipitation during spring and especially winter increases significantly, which might explain the absences of overall negative trends during this period. Additional to the precipitation increase in winter, air temperature increased during winter with a high acceleration during the last 20 years. Considering the mountainous environment, a decrease in snow contribution especially during the last 20 years can be assumed in coherence with previous studies showing a significant decrease in snow precipitations in the north

part of Jura Mountains (Charlier et al., 2022). Therefore, it can be summarized that long-term changes in the Jura Mountain are mainly related to increases in temperature, influencing snow contribution in the cold seasons and increasing evapotranspiration during the warm seasons. Both effects related to increasing temperature have been highlighted in previous studies of karst systems in temperate climate (Fan et al., 2023) but also mountainous regions in Mediterranean climate (Lorenzi et al., 2022). However, in the case of Jura mountains this does not lead to overall regional decreasing discharge. A possible

explanation is that increased evapotranspiration is compensated by higher cold-season precipitation during the last 20 years, a process already discussed for a case study in south-western England (Brenner et al., 2018).

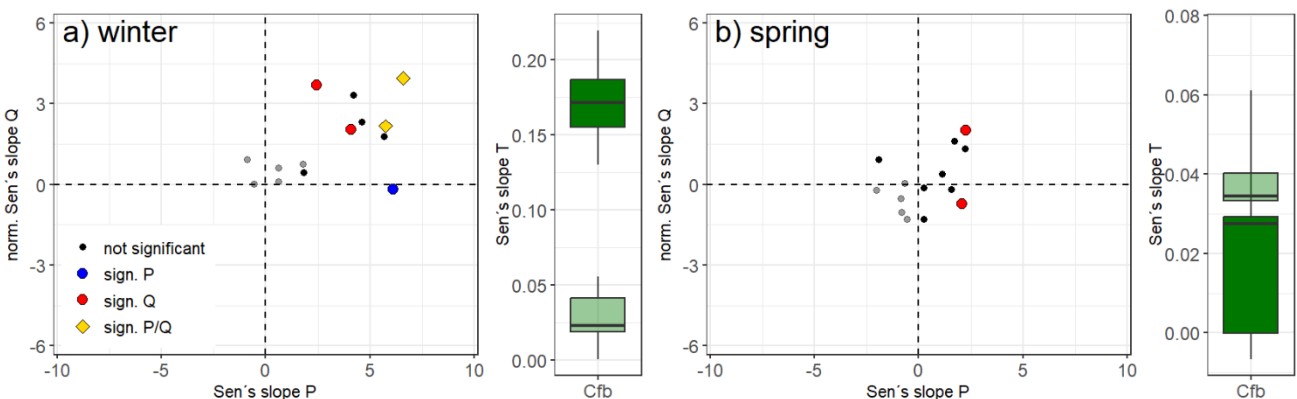



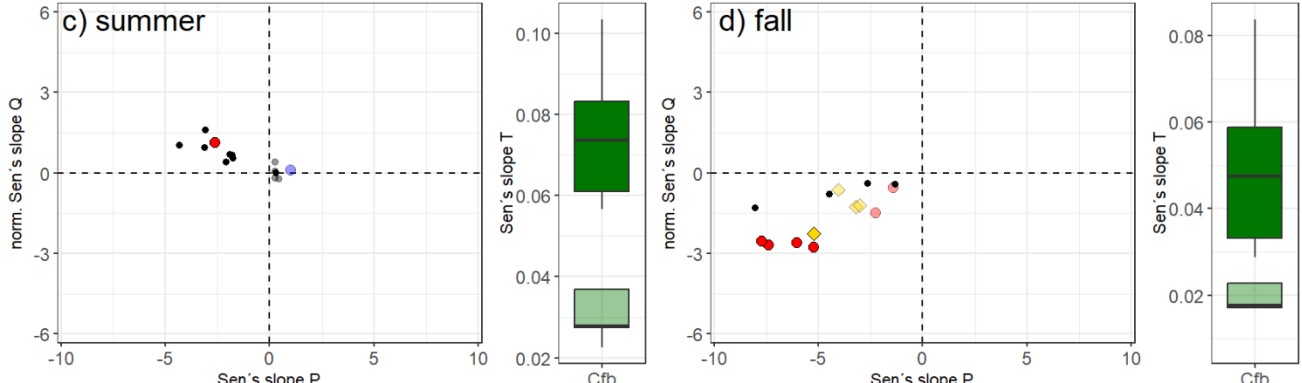

**Figure 6: Comparison of changes in precipitation and discharge trends for a) winter, b) spring, c) summer, and d) fall in combination**
**with temperature (boxplot) for all springs located in the Jura mountains (France). Springs indicated by red dots have significant**
**changes in discharge, by blue dots in precipitations and diamonds have significant changes in both seasonal precipitation and**
**discharge. Bold symbols represent the 20-year period.**

The Grand Causses region is of specific interest due to its location along a climate zone boundary. Previous analysis highlighted
that those areas or areas with changes in snow contribution are prone for changes in river discharge (Berghuijs et al., 2014)
and groundwater level variability (Nygren et al., 2020; Nygren et al., 2021). The climate boundary in the Grand Causses region
is connected to elevation differences, dividing the lower parts with Mediterranean climate from the parts with higher elevation
(temperate climate). Summarizing the results from Fig. 2 b), several springs in the region show positive discharge trends for
the last 20 years. However, precipitation increases mainly in spring and has a clear negative trend in fall and winter for both
climate zones (Fig.7), in link with a lower occurrence of Mediterranean storm events – that occurred in fall season - in the last
decades. The highest number of significant changes in discharge occurs in summer, all of them positive. Similar to the Jura
mountains, temperatures increase in all seasons, most pronounces in winter. Despite the decrease in winter precipitation, spring
discharge increases, even though not significantly. This regional increase in discharge can be seen in all seasons except from
fall. This last point is coherent with the decrease of precipitation, and thus of recharge during fall season. A possible explanation
resulting in overall increasing discharge, might be a strong reduction in snow contribution, leading together with increased
spring precipitation to a total saturation of the systems. Despite increased temperatures in the warm seasons, this leads to
increase discharge until fall.



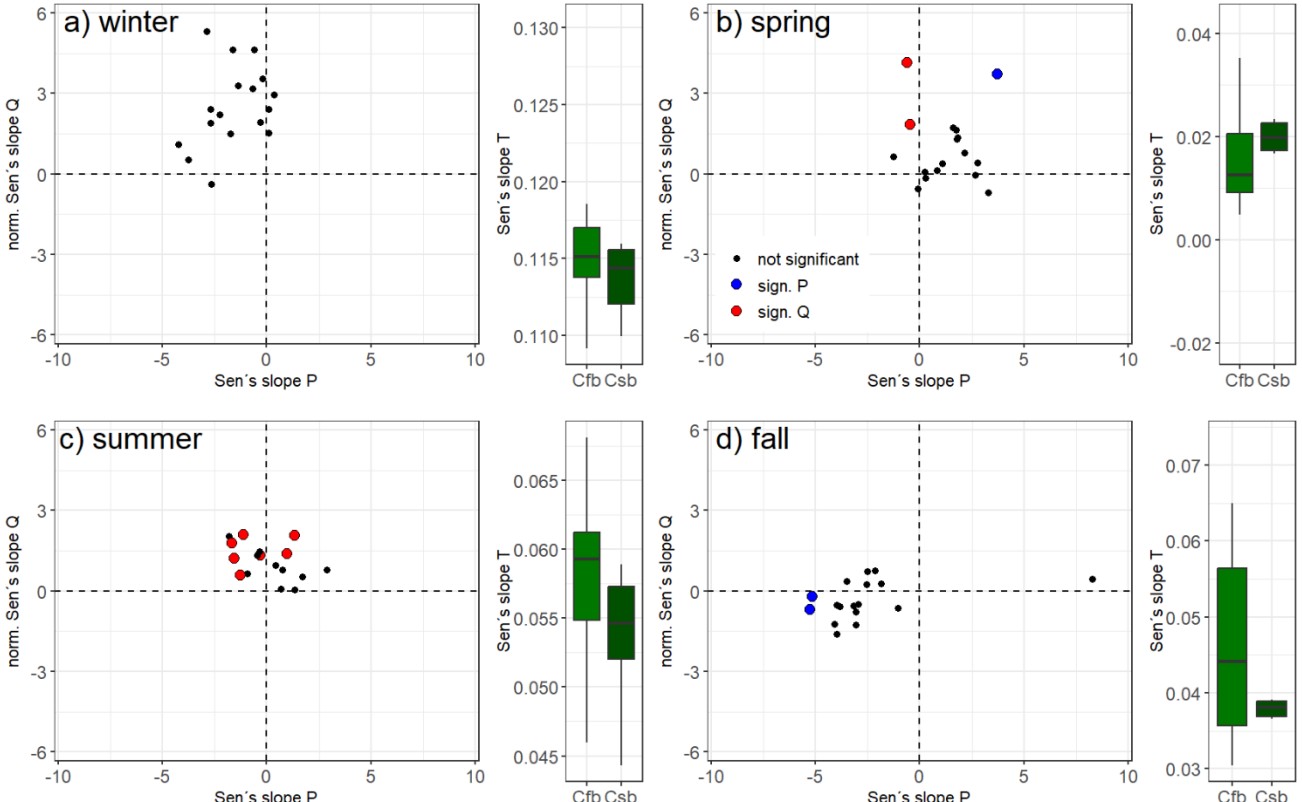

**Figure 7: Comparison of changes in precipitation and discharge trends for a) winter, b) spring, c) summer, and d) fall in combination with temperature (boxplot) for all springs located in the Grand Causses region (France). Springs indicated by red dots have significant changes in discharge, by blue dots in precipitations and diamonds have significant changes in both seasonal precipitation and discharge.**

## 4.3 Additional forcings impacting-karst water resources

Following the reasoning for the two French karst areas of the Jura Mountains and Grands Causses region in the previous section, it can be concluded that in absence of precipitation changes, temperature is the main driver of climate change related trends in European karst discharge. Anthropogenic warming is closely connected to increased evapotranspiration. This has an impact on the entire hydrological cycle and is closely connected to the occurrence of meteorological (e.g. Hänsel et al., 2019, Philip et al., 2020), soil moisture (e.g. Samaniego et al., 2018, Philip et al., 2020), and groundwater drought (e.g. Bloomfield et al. 2019).

Another factor influencing evapotranspiration is changes in land use or land cover. Several studies highlight the influence of land use changes on trends in river discharge (e.g. Vicente-Serrano et al., 2019) and extreme events, especially the spatial and temporal occurrence and severity of drought events (e.g. Brunner and Stahl, 2023). Even short-term changes due to crop rotations in agricultural areas have a direct impact on groundwater storage (e.g. Dakhlalla et al., 2016). Globally, land use or



land cover changes in karst areas could be detected for almost 5% of the surface for the period between 1992 and 2020, most of them related to agricultural reclamation or reforestation (Zhang et al., 2023). The impact of these changes is difficult to analyse on a larger scale since fundamental research on percolation and recharge processes is karst areas is lacking, especially for those covered by forest (Vilhar et al., 2022).

Possible explanations for the increase in discharge or moderation of downward trends - not only in the Grands Causses region - can also be found in the dependency of spring discharge on large-scale atmospheric circulations, as in Italian karst areas (e.g. Fiorillo et al., 2021). Sudden changes or even breaks in the system behaviour during the last decades can also be detected in other compartments of the hydrological cycle. These are closely connected to fluctuations of large-scale atmospheric circulations, which drive different hydroclimatic conditions (e.g. wind speed, temperature, precipitation) over Europe (e.g. Ionita et al., 2022; Deng et al., 2022). For European river basins, the 1980s highlight a starting point where periodic changes in river discharge closely connected to large-scale atmospheric circulations, e.g. the North Atlantic Oscillation (NAO), occur for the first time (Lorenzo-Lacruz et al., 2022). This transient connection between periodical fluctuations in European river discharge and large-scale atmospheric circulations can be detected on both continental (e.g. Lorenzo-Lacruz et al., 2022) and regional (e.g. Giuntoli et al., 2013; Boé and Habets, 2014) scales. Furthermore, the atmospheric circulation has an impact on long-term groundwater level variability (e.g. Holman et al., 2011; Neves et al., 2019; Rust et al., 2018; Rust et al., 2019, Baulon et al., 2022). These periodic signals propagate via recharge to the aquifer and account on average for 40 to 55 % of groundwater level variability and therefore play an as important role as current climate conditions (Neves et al., 2019; Rust et al., 2019).

## 5 Conclusion

Observational data from European karst areas were analysed for two different period. The first period focuses on European springs with the longest available time series and the second one covers the last two decades. This shorter period was chosen to identify any possible acceleration or moderation on a regional scale. Analysing trends in observational hydroclimatic and hydrological variables over the last 20 years provide a continental insight into these changes and at the same time allow for a higher number of karst springs to cover a wide range of hydraulic properties, climates, and topographies.

Although hydrological properties of karst systems differ widely, the results highlight the independency of long-term trends in discharge from the maturity or degree of karstification. Both, systems with a dominant fast flow component as well as systems with a dominant base flow component show significant increasing and decreasing discharge trends, respectively. Out of the two investigated hydroclimatic variables, the increase in temperature played a major role in the explanation of the discovered trends. The impact of changes in precipitation, including seasonal changes, is not able to explain the detected changes in discharge. The analysis of observation data simplifies groundwater recharge and flow processes in karst areas to a degree that system changes cannot be detected. As mentioned, the list of possible drivers of such changes is long so only a few of them were highlighted here. One important group are changing climatic conditions which can be divided into cyclic changes, due to large-scale atmospheric circulations, and continuous changes related to the timing and intensity of precipitation and



temperature changes during short-term periods, e.g. seasons. These types of changes can be considered by process-based groundwater recharge model approaches. Applying these types of models and analysing changes in discharge from karst areas with their unique hydrologic properties might overcome the disadvantage related to the validation of those models.

However, the here presented results have a practical relevance for modelling discharge in karst areas. Most time-series are rather short which bears the risk of using periods which are affected by changing climate conditions. Building up numerical models and validate them on trend-effected time series without considering the drivers or changing processes cause high risk of misleading future predictions of spring discharge. This also includes short-term fluctuations in climatic drivers, e.g. caused by large-scale atmospheric circulation.

**Author contribution**

Conceptualization: MG, JBC, YC, AH, Formal Analysis: MG, Funding acquisition: MG, JBC, YC, Methodology: MG, JBC, YC, AH, Software: MG, Visualization: MG, Writing – original draft preparation: MG, JBC, YC, AH, Writing – review & editing: MG, JBC, YC, AH

**Competing interests**

The authors declare that they have no conflict of interest.

**Financial support**

MG was financed within the MOPGA Fellowship Program under grant 127244W.

**Acknowledgments**

The Authors would like to thank the French Ministry of Europe and Foreign Affairs for its financial support within the
MOPGA Fellowship Program.

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
