# Peer review of "Trends in long-term hydrological data from European karst areas: insights for groundwater recharge evaluation"

_EGUsphere, 2024_

## Author Comment (AC3)

**CC1:**

*General comments*

*Excellent research in karst hydrology that just needs more detail before publication.*

**RESPONSE:** We thank the reviewer for his very positive comment.

*Line 13. Specify the European countries where you have studied the springs in the abstract.*

**RESPONSE:** We will consider this suggestion.

*Lines 25-88. The climate is affected by anthropogenic activities such input of $CO_2$. You analyse exclusively the last 20 – 40 years that register other types of climate variations with different driving forces. This scientific point is clarified and discussed exclusively later in the text and not in the introduction. Please, consider to modify the introduction.*

**RESPONSE:** As we mention in the response letter to reviewer 1, we will revise the introduction.

*Line 65. "Rising temperatures and the consequent reduction of snow contribution". Please, add more recent literature on snowmelt aquifer recharge in karst environments:*

*- Tracking flowpaths in a complex karst system through tracer test and hydrogeochemical monitoring: Implications for groundwater protection (Gran Sasso, Italy). Heliyon, 10(2).*

*- Snowmelt as a determinant factor in the hydrogeological behaviour of high mountain karst aquifers: The Garcés karst system, Central Pyrenees (Spain), Science of the Total Environment, 748, 141363*

**RESPONSE:** We will consider this suggestion and include some more literature after a brief literature review.

*Lines 86-89. Very good that you specify the objectives of your research using bulletin points*

**RESPONSE:** Thank you.

*Line 105. "Storativity" is clear, but not "inertia". Please, provide explanation for this concept/terminology in karst hydrology.*

**RESPONSE:** Inertia describes the karst system's resistance to rapid changes in response to external input (e.g. recharge). We will add a short explanation for this term in the main text.

*Line 428. "First period", specify four decades. You cannot recall the attention of the reader only for the for the second period.*

**RESPONSE:** We will add a specification to the first sentence of the conclusion.

*Line 462. Please, integrate recent papers on karst hydrology.*

 **RESPONSE: We will consider including more literature as a**

*General comments*

*Figure 1. The map can be much larger.*

**RESPONSE:** We agree and make changes accordingly.

*Figures 2, 4 and 5. Remove the underscore from the legend.*

**RESPONSE:** Thank you for this observation. We will change it.

*Figures 6 and 7. Please, provide more detail on the box plots in the captions. The red line? Median, or arithmetic average?*

**RESPONSE:** The line in the boxplot represents the median ($50^{th}$ percentile). We will add this information to the captions of Figure 6 and 7.

---

## Author Response (AR1)

**RC1:**

*The manuscript entitled "Trends in long-term hydrological data from European karst areas: insights for groundwater recharge evaluation" presents an analysis of daily karst spring discharges across Europe and aims to identify connections between daily climate variables and karst discharge. I find this topic very interesting, and HESS is a suitable journal for this type of study. However, the manuscript has several weaknesses, and in my opinion, several major and minor revisions are necessary before considering this manuscript for publication.*

**RESPONSE:** The authors would like to thank the reviewer for the positive feedback on our work and the insightful comments which we address in the following.

*To avoid confusion, it would be helpful to clearly differentiate between recharge, actual recharge (mentioned in line 32), and absolute GW recharge (line 80). What are the differences between these terms?*

**RESPONSE:** Groundwater recharge is used to describe the process. The terms actual recharge and potential recharge are used to distinguish between recharge estimates based on saturated-zone and unsaturated-zone methods, respectively. We will keep the term actual recharge. 'Absolute groundwater recharge' (L80) was initially used to refer to recharge values given in absolute terms, as opposed to percentage-based recharge. However, since no other recharge-related numerical values from the cited publication were included, we will remove the term 'absolute' and use only groundwater recharge.

***Changes:*** *We did all the changes according to our answer to the reviewer.*

*In line 87: The phrasing of this question is unclear. What do you mean by "significant change"? What does "overall spring discharge" refer to? Are you discussing low and high flow conditions of what specifically?*

**RESPONSE:** Significant change means an increase or decrease according to the Mann-Kendall test. This is further explained in L174 where it states: 'To overcome the sensitivity of the confidence level of the Mann-Kendall test, two different statistical significance levels were set, the first one with p-values of 5 %, and the second one with p-values of 10 %.' Overall spring discharge' refers to the changes in measured spring discharge over the two periods. However, we agree that we can simplify the terminology and refer to it as spring discharge. Low and high flow conditions are further defined as '0.1 and 0.9 percentile of spring discharge' (L246).

*Changes: We changed the research question to:* Does variability in discharge have an impact on low and high flow conditions? *(L89)*

*In line 88: To which karst storage properties are you referring? How can a property be sensitive to climate change?*

**RESPONSE:** We agree with the reviewer that this is a misleading phrasing and suggest a change to: 'Is it possible to identify karst units with certain properties related to storage which are particularly sensitive to climate change? '

*Changes: We changed the research question to:* Is it possible to identify karst areas with certain properties related to storage which are particularly sensitive to climate change?

*In its current form, the introduction does not clearly convey the study's objectives. Please restructure the introduction to improve readability and ensure it includes all the necessary information to understand the objectives. Additionally, it is important to cite studies that have conducted similar research, so that you can clearly highlight the novel aspects of your study and conclusions.*

**RESPONSE:** We will revise the introduction. Regarding your additional point, the introduction includes an overview of the available literature. To our knowledge, there is a lack of regional studies focusing on discharge changes connected to hydroclimatic drivers, and we are not aware of any such analysis in a continental context. These are the reasons we consider this research novel.

*Changes: We revised the introduction and added text describing the functioning of karst systems, e.g.* Hydraulic properties such as permeability, porosity, and storativity govern the time-response of these systems. The storativity of a karst aquifer, which refers to the ability of the aquifer to store and release water, plays a crucial role in the response of spring discharge to recharge events. The interaction between conduit and matrix flow components introduces a time lag in the aquifer's response to recharge referred to as the system's inertia. *(L52) While we agree on the importance of citing similar studies to contextualize our research, a thorough review of the literature has not revealed any prior studies that specifically address this particular aspect of our topic.*

*Line 125: Why is it important to mention that some of the investigated areas lie along climate zone boundaries? This detail seems disconnected from the rest of the paragraph. Why is this not reflected in Table 1?*

**RESPONSE:** We do not think that this is disconnected from the rest of the paragraph, where we discuss the location of the springs and introduce the different climate zones. This is important to mention because climate zones are not stable over time, and, as we later

explain, regions along climate zone boundaries are 'prone to changes in river discharge (Berghuijs et al., 2014) and groundwater level variability (Nygren et al., 2020; Nygren et al., 2021). ' The climate zone, as defined in the Table caption, represents the location according to the Köppen-Geiger classification (1986-2010). In this classification, sharp boundaries exist between the different climate zones.

***Changes:*** *No changes*

*Line 145: You mention that you used the R package BFI. Why did you choose this method over others? How does this package separate quick from slow flow components? How does the package calculate the BFI? How much do the results and conclusions of your study change if the value of alpha is different?*

**RESPONSE:** The BFI package calculates the BFI according to the principles described by Line and Hollick (1979). We will briefly summarize the process: the method is based on a digital filter that works iteratively, applying a smoothing algorithm to reduce peaks in the hydrograph (quick flow component). The filter coefficient $\alpha$ influences the degree of smoothing and sets a threshold to separate high-frequency (quick) flows from low-frequency (slow, baseflow) components. Once the time series is split into quick flow and baseflow components, the BFI is calculated as the ratio between baseflow and total discharge. In general, lower values indicate a system dominated by quick flow, where streamflow is more responsive to precipitation events.

The predefined values for the filter coefficient have only a minor impact on the results and conclusions of this study. The base-flow index is used only for a simplified classification of the springs, and the exact location is of secondary importance. Normally, karst spring classification based on spring discharge is done by analyzing single events (e.g., Bailly-Comte et al., 2023), but the temporal and spatial extent of the analysis presented in our paper does not allow for such an in-depth analysis.

***Changes:*** *We added some more information on the BFI calculation:* This method can be described as a smoothing algorithm using a one-parameter recursive digital filter. The filter coefficient $\alpha$ influences the degree of smoothing and sets a threshold to separate high-frequency (quick) flows from low-frequency (slow, baseflow) components. *(L150) and* After splitting the time series into quick flow and baseflow components, the BFI is calculated as the ratio between baseflow and total discharge. *(L155)*

*Could you explain in Section 2.3 the meaning of a positive and negative Sen Slope? How are these slopes calculated?*

**RESPONSE:** We will add a short description about the meaning of Sen´s slope and one sentence to explain the meaning of Sen´s slopes.

*Changes: We revised the section 2.3 and added the required information:* In addition, Sen's slope (Sen, 1968) was calculated to determine the magnitude and direction of the trend, with positive values indicating increasing trends and negative values indicating decreasing trends. *(L186)*

*Line 160: Why do you analyze the information from only one cell? Is it possible that karst capture areas are larger in extent than 0.1 degrees (about 10 kilometers)? If so, what are the implications of this for your study?*

**RESPONSE:** Unlike river catchments, it is difficult to delineate karst catchments (see, for example, Geyer et al., 2013). Catchment limits may change depending on the characteristics of the karst system, local topography, and hydrogeological connectivity. Large karst aquifers often span several kilometers and may not align with surface topography, meaning that a single 0.1° cell may not fully capture the entire recharge area. We are aware that this might impact the results for large systems (i.e., those larger than a single cell area) and will mention this in the text.

*Changes: We added a sentence to address the problem:* However, for karst springs with catchments larger than the cell size or high topographical gradient, this approach might not fully capture local variability. *(L168)*

*Line 177: With your data, you could evaluate how close or far your discharge distributions are from normal. I would recommend calculating this instead of simply assuming that your data is not normal.*

**RESPONSE:** The Mann-Kendall test does not require the distribution to be normal, so we think that the normal condition does not need to be tested in the context of the paper.

*Changes: No changes*

*Line 183: You initially mentioned that 6 springs exhibit significant changes. Then, explained that 3 of them exhibit negative and 2 positive variation. What kind of variation does the remaining spring exhibit—positive or negative?*

**RESPONSE:** Thank you for pointing this out. We will change the text.

*Changes:  We double-checked the data and realized that the text was indeed correct. There are two springs in the French Jura Mountains with a negative trend over 40 years (located really close to each other). Here is a copy of the text where we highlight (bold) the six springs:* Of the exceptions, one spring is situated in the Mediterranean climate zone (Csa), and

the other in the humid subtropical climate zone (Cfa). **Both show decreasing spring discharge**. The latter spring (Fontestorbes) drains a mountainous catchment in the Pyrenees. **The other two springs with decreasing spring discharge** (Lison and Verneau) are located in the French Jura mountains, which is also a mountainous region. In contrast, the Spring Pissling Ursprung, which drains a karst system on the northern slopes of the Austrian Alps, shows an increasing discharge trend. **It is one of the two springs with positive trends, the other being Spring Cheriton in England.**

We revised this part to make it clear for the reader.

*Why do you include figure 2a if you never mention it in the text?*

**RESPONSE:** We will mention it in the text.

***Changes:*** *Added in L194*

*Line 222: How can you justify this assumption? To me, it is critical that you disregard the effects of temperature variability when you are trying to quantify the impact of climate variables on karst spring discharge.*

**RESPONSE:** We do not ignore the effect of temperature variability. We state that we do not consider temperature in the following analysis because Fig. 2f clearly shows an overall increase in temperature across mainland Europe, which does not account for the observed increase in spring discharge over the same period (Fig. 2b). The main part of the paper focuses on trend analysis, which differs from variability. There are distinct differences between these two terms, such as timescales (long-term vs. short-term), patterns (directional vs. fluctuations around a mean), and implications (potential shifts in climate/environmental conditions vs. natural cycles/extremes). Nonetheless, we included an analysis of seasonal temperature trends in Section 4.2 and, for example, conclude that 'it can be summarized that long-term changes in the Jura Mountains are mainly related to increases in temperature, influencing snow contribution in the cold seasons and increasing evapotranspiration during the warm seasons' (L365).

***Changes:*** *No changes*

*The label for Figure 4 refers to subplots a and b, but only one figure with an inset is presented. Following Section 3.3 is complicated; it is unclear which figure corresponds to the 20-year period and which to the 40-year period*

**RESPONSE:** The large figure corresponds to the 20-year period and the inset to the 40-year period. We will add labels to the figures.

***Changes:*** *We revised all figures and included a label for Figure 4b.*

*You begin the discussion section by arguing that precipitation trends are rare and hardly explain variability in karst discharge. However, I think the temporal scale should be considered. Daily precipitation trends might be rare, but hourly precipitation trends could be common and significant. Could you elaborate on this?*

**RESPONSE:** Yes, we acknowledge that we state there are hardly any significant trends in precipitation for both periods, as shown in Fig. 2c and 2d. We also agree that the temporal scale must be considered; hourly precipitation trends may be common, and significant for discharge patterns in the areas investigated, although we do not see a direct connection to the long-term trend analysis presented here. We also noted this in the discussion. For example, 'but due to the fact that trends in spring discharge might not only be influenced by long-term changes in hydroclimatic conditions but also short-term (e.g., seasonal) changes in processes related to groundwater recharge and storage' (L300), and 'This strongly indicates changes in the partitioning between concentrated and diffuse recharge, suggesting changes in precipitation patterns' (L342).

***Changes:*** *No changes*

*In Section 4.1, you mention that karst spring flow follows a similar pattern to that depicted by streamflow. If precipitation variability is not important, how can you justify this?*

**RESPONSE:** We respectfully disagree with this comment. At no point in the manuscript do we state that 'precipitation variability is not important,' as we do not analyze precipitation variability. This work focuses on long-term (seasonal) trends in karst spring discharge and hydroclimatic conditions. However, our findings even led us to conclude that 'This is a strong indication of changes in the partitioning between concentrated and diffuse recharge, and therefore suggests changes in precipitation patterns' (L342), implicitly incorporating precipitation variability

***Changes:*** *No changes*

*Line 27: global change of what? Climate change?*

**RESPONSE:** Global change is a commonly used expression including climate change but also human societies and the impact of their activities. We refer to literature.

***Changes:*** *No changes*

*Line 32: Is it possible to directly measure integrated (over space) recharge at regional scale?*

**RESPONSE:** No, direct measurement of spatially integrated recharge at a regional scale is not feasible.

***Changes:*** *No changes*

*Line 51: Why GW recharge "needs" to be divided into concentrated and diffuse processes?*

**RESPONSE:** Groundwater recharge in general does not need to be divided into concentrated and diffuse processes, but it is necessary for conceptual modelling of karst systems. This is a direct consequence of the influence of karst systems properties on groundwater flows, described extensively in literature. We refer to the literature mentioned in line 50 to 54.

***Changes:*** *We revised this part of the introduction and added a few more information.*

*Line 58: Efficient in which sense?*

**RESPONSE:** We will delete "efficient" and replace it with "important".

***Changes****: Replaced: important*

*Line 73: Correlated to what?*

**RESPONSE:** We deleted "correlated to".

***Changes****: According to our answer.*

*Line 75: Which authors?*

**RESPONSE:** The authors of the last-mentioned publication, in this case De Vita et al., 2012.

***Changes****: Changed to: According to the latter…*

*Line 84: Multi-decadal*

**RESPONSE:** Changed according to the comment.

***Changes:*** Changed according to the comment.

*Line 100: Which quantiles?*

**RESPONSE:** 0.1 and 0.9 percentile

***Changes:*** *We added this information in L105:* Changes were evaluated using the Mann-Kendall test and Sen's slope computation (Section 2.3) for both monthly spring discharge and annual extremes, expressed as different quantiles, namely the 10th and 90th percentile.

*Line 112: How is this information combined?*

**RESPONSE:** A detailed description of the WOKAS data set can be found in Olarinoye et al. (2020). Time series from both databases, WOKAS and Hydroportail, that met the requirements were combined into a single database, ensuring duplicates were avoided. .

***Changes****: No changes*

*Lines 118-122: Check correct use of word "respectively". Do this in the rest of the document too.*

**RESPONSE:** We will do it and make changes accordingly.

***Changes:*** *We did a thorough language check on the entire manuscript.*

*Line 118: It is written "No further pre-processing was done prior to the analysis". It is not clear what preprocessing you applied to the data. Do you refer to the data selection?*

**RESPONSE:** We agree that the sentence is as misleading. We have not done any pre-processing and therefore delete "further" from the sentence.

***Changes:*** According to the comment.

*Also, you mentioned that data is available for four countries. Maybe it is a good idea to write their names, given that in Table 1 you only list abbreviations.*

**RESPONSE:** We will add an explanation of these official abbreviations following the ISO 3166-1 (alpha 2 code) to the table caption of Table 1.

***Changes: We added*** Country codes: Austria (AT), France (FR), United Kingdom of Great Britain (GB), Slovenia (SI) ***to the table caption.***

*Line 169: Why the spring discharge?*

**RESPONSE:** Without further context it is not possible to understand what the reviewer wants to point out.

***Changes****: No changes*

*Line 232: How many? Could you add this information to the figure?*

**RESPONSE:** It is not entirely clear what kind of information reviewer 1 is asking for. Every point in Figure 3 represents one of the springs resulting in 54 data points for the short period and 22 data points for the long period.

*Changes: No changes*

*Figure 1 label: I see orange symbols.*

**RESPONSE:** There are no orange symbols in Figure 1, the reviewer might refer to Figure 2. The colors used in the figures follow the recommendations for a better visibility for people with color vision deficiency.

*Changes: As already mentioned, we revised all figures.*

**RC2:**

*In this study, the authors use observational data of precipitation, temperature, and discharge of more than 50 karst springs spatially distributed over Europe to give a continental overview of changes in groundwater resources in karst areas. They perform a trend analysis using Mann-Kendall and Sen's slope on two different periods of 20 and 40 years, stating that long-term trends in spring discharge follow the general pattern of river discharge found in literature, while the last 20 years deviate from this behavior, mainly influenced by the temperature increase. Possible process changes were assessed by analyzing also trends in high and low flow values, and structural differences of the systems were considered by using two indices related to the storage and inertia of the system. The results of the analysis of the observed trend of hydroclimatic and hydrological variables are discussed with respect to the indirect influence of other drivers such as changes in land use or land cover, specific regional conditions but also changes in processes related to groundwater recharge and storage, providing insights for assessing groundwater recharge in the past and in the future.*

*The scientific contribution of this paper falls within the scope of Hydrology and Earth System Sciences. The results are discussed in an appropriate and balanced way; the paper is well-written with a clear and well-organized structure.*

**RESPONSE:** We thank the reviewer for contributing his comments and highly appreciate the overall positive assessment of our work.

*A more mathematical and detailed explanation of how the Mann-Kendall test and Sen's slope are performed could be included in paragraph 2.3 to help the reader better interpret the results of the study.*

**RESPONSE:** We will add a more detailed explanation on the Mann-Kendall test and Sen's slope.

***Changes:*** *We revised the section 2.3 and added a more detailed explanation of the Mann-Kendall test and Sen's slope, e.g.* Specifically, the test statistic is computed as the sum of the signs of differences between all pairs of data points, with the variance adjusted using a variance inflation factor derived from the autocorrelation structure. For a detailed explanation, readers are referred to Hamed and Rao (1998). *(L185)*

*The final connection to modeling approaches provided in the conclusions is not clear: can you explain better how these results impact the discharge modeling?*

**RESPONSE:** While it does not have a direct impact on discharge modeling, it is something that should be considered when predicting future spring discharge. As stated in the conclusions, most time series of karst discharge are relatively short, and therefore there is a high risk that these time series are influenced by trends. If we use these time series without accounting for the trends and their underlying drivers in future predictions, there is a high risk of misrepresenting future conditions, assuming these trends will continue linearly and are representative of future system behaviors.

***Changes:*** *We addressed the comment by changing the last paragraph:* However, the here presented results have a practical implication for modelling discharge in karst areas. Most time series are relatively short, which increases the risk of basing analyses on periods influenced by changing climate conditions without taking account the associated shifts. Moreover, developing numerical models and validating them using trend-affected time series—without accounting for the underlying drivers or evolving processes—risks producing misleading future predictions of spring discharge by assuming continuous linear trends. *(L458)*

*The visualization of the results is crucial and I found all the figures suitable to convey the different messages about trends, changes, and relationships in the observed variables. However, I suggest improving the selection of the markers to make the plots more effective, clear, and straightforward to interpret. In particular, I have difficulties identifying bold symbols.*

**RESPONSE:** We already tried different ways of visualizing the results in a comprehensive manner but understand the difficulties mentioned by the reviewer. We will find a better way to distinguish between the two different periods.

*Changes:* All Figures and legends are remade to improve the clarity.

*Line 162: clarify the procedure reformulating the sentence "Daily values of precipitation and temperature were accumulated respective averaged to obtain monthly and seasonal values."*

**RESPONSE:** We agree that the description lacks some basic information, which will be added to the text.

*Changes:  We changed the sentence to:* For trends analysis, daily precipitation values were summed, and daily temperature values were averaged to calculate monthly and seasonal values. *(L172)*

*Fig. 4: the caption refers to panels a) and b) but in the figure no label is provided to identify the panels.*

**RESPONSE**: We will include the labels.

*Changes:  As already mentioned, we revised all figures and included a label for Figure 4b.*

**CC1:**

*General comments*

*Excellent research in karst hydrology that just needs more detail before publication.*

 **RESPONSE:** We thank the reviewer for his very positive comment.

*Line 13. Specify the European countries where you have studied the springs in the abstract.*

**RESPONSE:** We will consider this suggestion.

*Changes: We added the required information:* This study analyses observational data (precipitation, temperature and discharge) from over 50 springs distributed across Europe (AT, FR, GB, SI), offering a continental perspective on groundwater resource changes in karst areas. *(L12)*

*Lines 25-88. The climate is affected by anthropogenic activities such input of CO2. You analyse exclusively the last 20 – 40 years that register other types of climate variations with different driving forces. This scientific point is clarified and discussed exclusively later in the text and not in the introduction. Please, consider to modify the introduction.*

**RESPONSE:** As we mention in the response letter to reviewer 1, we will revise the introduction.

***Changes:*** *We revised the introduction however we think that other aspects are already discussed in the second part of the introduction - L71 and following*

*Line 65. "Rising temperatures and the consequent reduction of snow contribution". Please, add more recent literature on snowmelt aquifer recharge in karst environments:*

*- Tracking flowpaths in a complex karst system through tracer test and hydrogeochemical monitoring: Implications for groundwater protection (Gran Sasso, Italy). Heliyon, 10(2).*

*- Snowmelt as a determinant factor in the hydrogeological behaviour of high mountain karst aquifers: The Garcés karst system, Central Pyrenees (Spain), Science of the Total Environment, 748, 141363*

**RESPONSE:** We will consider this suggestion and include some more literature after a brief literature review.

***Changes:*** *We added in total three more papers discussing snow melt contribution.*

*Lines 86-89. Very good that you specify the objectives of your research using bulletin points*

**RESPONSE:** Thank you.

***Changes:*** *No changes*

*Line 105. "Storativity" is clear, but not "inertia". Please, provide explanation for this concept/terminology in karst hydrology.*

**RESPONSE:** Inertia describes the karst system's resistance to rapid changes in response to external input (e.g. recharge). We will add a short explanation for this term in the main text.

***Changes:*** We added a short explanation of the term: The interaction between conduit and matrix flow components introduces a time lag in the aquifer's response to recharge referred to as the system's inertia. (L54)

*Line 428. "First period", specify four decades. You cannot recall the attention of the reader only for the for the second period.*

**Changes:** *Changed according to the comment.*

*Line 462. Please, integrate recent papers on karst hydrology.*

 **RESPONSE: We will consider including more literature as a**

**Changes:** *We added a few more papers.*

*General comments*

*Figure 1. The map can be much larger.*

**RESPONSE:** We agree and make changes accordingly.

**Changes:** *Done but anyways the map is scalable.*

*Figures 2, 4 and 5. Remove the underscore from the legend.*

**RESPONSE:** Thank you for this observation. We will change it.

**Changes:** *We revised all figures and legends*

*Figures 6 and 7. Please, provide more detail on the box plots in the captions. The red line? Median, or arithmetic average?*

**RESPONSE:** The line in the boxplot represents the median (50$^{th}$ percentile). We will add this information to the captions of Figure 6 and 7.

**Changes:** *We added the information to the captions.*